# Pancreatic Cancer Cells Induce MicroRNA Deregulation in Platelets

**DOI:** 10.3390/ijms231911438

**Published:** 2022-09-28

**Authors:** Jorge Yassen Díaz-Blancas, Ismael Dominguez-Rosado, Carlos Chan-Nuñez, Jorge Melendez-Zajgla, Vilma Maldonado

**Affiliations:** 1Posgrado en Ciencias Biológicas, UNAM, Mexico City 04510, Mexico; 2Epigenetics Laboratory (2), Instituto Nacional de Medicina Genómica (INMEGEN), Mexico City 14160, Mexico; 3Department of Surgery, National Institute of Health Sciences and Nutrition Salvador Zubirán, Mexico City 14080, Mexico; 4Functional Cancer Genomics Laboratory, Instituto Nacional de Medicina Genómica (INMEGEN), Mexico City 14160, Mexico

**Keywords:** miRNAs, pancreas cancer, tumor-educated platelets

## Abstract

Pancreatic cancer is a pathology with a high mortality rate since it is detected at advanced stages, so the search for early-stage diagnostic biomarkers is essential. Liquid biopsies are currently being explored for this purpose and educated platelets are a good candidate, since they are known to present a bidirectional interaction with tumor cells. In this work, we analyzed the effects of platelets on cancer cells’ viability, as determined by MTT, migration using transwell assays, clonogenicity in soft agar and stemness by dilution assays and stem markers’ expression. We found that the co-culture of platelets and pancreatic cancer cells increased the proliferation and migration capacity of BXCP3 cells, augmented clonogenicity and induced higher levels of Nanog, Sox2 and Oct4 expression. As platelets can provide horizontal transfer of microRNAs, we also determined the differential expression of miRNAs in platelets obtained from a small cohort of pancreatic cancer patients and healthy subjects. We found clear differences in the expression of several miRNAs between platelets of patients with cancer healthy subjects. Moreover, when we analyzed microRNAs from the platelets of the pancreatic juice and blood derived from each of the cancer patients, interestingly we find differences between the blood- and pancreatic juice-derived platelets suggesting the presence of different subpopulations of platelets in cancer patients, which warrant further analysis.

## 1. Introduction

Pancreatic cancer represents one of the deadliest types of cancer. Pancreatic ductal adenocarcinoma (PDAC) accounts for >90% of all pancreatic cancer [1] and its mortality rate (4.4 per 100,000 cases) is almost as high as its incidence (4.8 per 100,000 cases) [2]. The only effective therapy against pancreatic cancer is the surgical removal of the whole affected area of the pancreas [3]; nevertheless, only around 15% of patients are eligible for this treatment at the time of diagnosis because of the advanced stage of the disease at diagnosis [4,5].

The late diagnosis in PDAC can be explained by the asymptomatic course in the earlier stages of this disease [6]. Paradoxically, PDAC is a type that develops very slowly, taking as long as 7–10 years from a localized lesion to a metastatic state [7]. This period between the early stages and metastatic disease represents an important opportunity to improve the overall survival for patients. According to the American Cancer Association (ACA 2022), if diagnosed in the earlier stages the 5-year survival of pancreatic cancer patients can climb up to 40%, and 10% of these patients can be disease-free after 5 years: nonetheless, the lack of specialized methodology in screening procedures represents the biggest obstacle for early diagnosis in the field [8,9,10].

Liquid biopsies could fill in the gap for screening procedures in the earlier detection of pancreatic cancer in patients [10]. Pancreatic cancer liquid biomarkers, such as circulating tumor cells or cell-free DNA, have been previously explored [4,11]; nonetheless, these markers are scarce and difficult to obtain or have low sensitivity [12]. To solve this problem some authors, such as Best et al. [13] have proposed the use of platelets as the biological source for liquid biopsies.

Platelets are small non-nucleated cells produced by megakaryocytes that are involved in various physio and pathological processes, such as coagulation [14], inflammation [15], immune response [16,17] and production of extracellular vesicles [16,17,18,19]. An association of platelets and cancer was first postulated by Trousseau in 1868; he found a “hypercoagulative state” in cancer patients [20]. As for platelets and pancreatic cancer, back in 1999 Schwarz established the platelet count as an independent survival marker where patients with thrombocytosis presented the worst survival rates. Recently, a meta-analysis conducted by Chen, Na, and Jian, with a total of 1756 patients in 13 cohorts, found the same predictive factor for thrombocytosis in pancreatic cancer [21]. Additionally, in recent years, platelets were found to be more complex than originally thought, playing a highly dynamic role in the tumor microenvironment [1]. Despite being non-nucleated, platelets can regulate the transcription of its pre-mRNA load (inherited from the megakaryocyte from which it derived) [22]. Platelets manage this differential expression through regulatory miRNAs [23,24].

MicroRNAs are short non-coding RNA species of 18 to 24 nucleotides that play a fundamental role in post-transcriptional gene regulation, silencing mRNA expression by coupling to their 3′ UTR, marking it for degradation [22,25]. Platelets can change their expressions of miRNAs’ load by external or internal stimulus or even through direct uptake from the microenvironment [11,26]. This plasticity makes the platelets a fundamental piece for any cellular microenvironment, even tumoral [16,27]. Platelets contribute to the tumoral microenvironment as suppliers of cytokines and other biological active molecules and as a physical shield from shear forces and immune cells [13,28,29]. More importantly, platelets transfer horizontal microRNAs using microvesicles as the carriers [30]. These microRNAs in turn can influence cancer and non-cancer cells in the tumoral microenvironment [26,31]. In 2015, Best et al. [32] demonstrated that platelets have differences in their RNA load depending on the type of tumor, which could be due to a process termed “education” and hypothesized about using these “tumor educated platelets” (TEPs) as markers for the health state of diverse organs [22]. More recently, it has been observed that during this education process, the platelets can also sequester tumor-specific biomolecules, such as miRNAs [20].

Platelet education is complex process that is not yet fully understood in pancreatic cancer. In 1995, Heinmoller et al., reported that the pancreatic carcinoma cells are able to induce platelet aggregation via activation of thrombin [33]. In 2017, Elaskalani et al. reported that platelet-derived factors could promote the proliferation of BxPC−3 and AsPC-1 cancer cells [34]. This was accompanied by a protective effect toward gentamicin cytotoxicity by an increase of Slug, a transcriptional factor known to be upregulated during cancer epithelial–mesenchymal transition. That same year, Mitrugno et al. published that pancreatic cancer cell proliferation was increased by human platelets in a c-Myc-dependent manner [35].

This work delves for the first time into the effect of platelets on pancreatic cancer cells, using a co-culture in vitro model.

## 2. Results

### 2.1. Exposition of MEG-01 Cell Line with Valproic Acid Produces Functional Platelets In Vitro

We used an in vitro model of platelet production to have a constant and reproducible supply of platelets using the megakaryoblast human cell line MEG−01, which has been extensively employed for this end [36]. Under basal conditions (RPMI medium +10% BFS), this cell line produces platelets at a low but steady rate. It has been previously demonstrated that the main physiologic megakaryocyte differentiation factor, thrombopoietin or valproic acid, induce the differentiation of MEG−01 cells to produce a large number of platelets [37,38]. We tested three conditions: exposure to 2 mM of valproic acid, 100 ng/mL of thrombopoietin (TPO), or both. We found that the addition of valproic acid alone for 72 h prior to collection had the best and more consistent results yielding as much as four times more (≈40,000 platelets/µL) than control (≈10,000 platelet/µL) (Figure 1A).

To corroborate that the MEG−01 derived platelets were functional, we performed clot formation assays by adding CaCl_2_ to collected platelets, as described previously [39]. As expected, platelets formed similar thrombi once CaCl_2_ was added (inset in Figure 1A).

We then conducted flow cytometry assays to assess the presence of the surface marker CD41 in the platelet-like particles present in the MEG−01 cultures and those derived from healthy donors. Figure 1B shows the histogram of platelets obtained from donor’s blood and from the MEG−01 differentiated cell line. These results show that MEG-01 were able to generate a relatively large number of functional platelets.

### 2.2. Platelet Effects on Cancer Cells

We then sought to evaluate the effect of the platelets on pancreatic cancer cells. For this, we co-cultured the pancreatic cancer cell line BxPC−3 with platelets derived from differentiated MEG-01 cells. In Figure 1C we show a representative clear field of these co-cultures, with an immunofluorescence validation in Figure 1D. CD41 platelet marker was visualized with an AlexaFlour488-coupled antibody (in green), whereas cellular membranes (BxPC−3 cells) were stained with the FM 4-64 dye (red). Nuclei were stained with DAPI (blue). In order to analyze the effects of platelets on pancreatic cancer cells, we measured the growth of BxPC−3 cells that were co-cultured with platelets. As shown in Figure 2A, we found an increased number of cells after exposure to platelets, reaching high significance after 28 h. We then tested migration of these cells using transwell inserts. The BxPC−3 cells that had been in contact with the platelets showed an increased invasion ability phenotype (Figure 2B,C). Finally, we measured the clonogenic ability. As shown in Figure 2D,E, we found an increase in the clonogenic capacity of the cells that were co-cultivated with platelets with two different methods. All of these results show that the interaction of pancreatic cancer cells with platelets confers a more aggressive phenotype.

### 2.3. BxPC-3 Cells in Direct Contact with Platelets Presented Stem Characteristics and Expressed Genes Associated to Stemness

Since pancreatic cancer cells have two interchangeable phenotypes (basal-like and classical) that vary in their stem capacity [41,42], and we found an increase in the clonogenic ability of BxPC−3 cells after exposure to platelets, we sought to analyze if these cells presented an increased stem capacity. For this, we performed an extreme limiting dilution analysis (ELDA) [43] on the cells exposed to platelets. Nine serial dilutions of BXPC−3 cells were used for the assay. As shown in Figure 3A–D, we found a 2.4-fold increase in the stem-like fraction in cells co-cultured with platelets. We then collected the BxPC−3 cells exposed to platelets and measured the expression of stem-related genes. As shown in Figure 3E, we found that the Bx-PC−3 cells that were co-cultured with platelets overexpressed NANOG, OCT4 and SOX2 genes (Figure 3E). These three genes are involved in self-renewal programs and were associated with a more stem-like phenotype in cancer cells.

### 2.4. Differential MicroRNAs Expression in Platelets Derived from Patients

Since platelets are non-nucleated, they do not have the ability to regulate gene expression via transcription, so they rely on post-transcriptional and post-translational mechanisms, such as microRNA-mediated mRNA decay and translational control. In addition, it has been shown that platelets not only produce vesicles that are able to transfer microRNAs and other cellular components to adjacent cells but that this transfer modulates neighboring cell functions [16]. For these reasons, we sought to analyze if platelets derived from the blood or pancreatic juices of patients with pancreatic adenocarcinoma (Appendix A) differed in their microRNA expression. As shown in Appendix A, we found an increase in the number of platelets in patients with cancer, as compared to healthy subjects [44]. These results are in accordance with previously reported results in pancreatic cancer where thrombocytosis was observed associated with poor survival [44]. Our results show that this increase in platelet count is more pronounced in patients with solid pseudo papillary tumor (SPPT) and villous adenoma (VA) (Appendix A).

We were able to obtain enough RNAs in both the blood platelets and pancreatic juice platelets of a small number of patients to perform a microRNA expression analysis. Figure 4A shows the Volcano plot graph of the differentially expressed microRNAs comparing the platelets derived from the blood of pancreatic cancer and healthy subjects. Interestingly, we found that a Principal Component Analysis (PCA) showed a clear separation of platelets’ microRNA expression between the blood of the cancer samples (blue circles) and those of the cancer, in which two groups are apparent: microRNAs derived from blood platelets (red circles) and those derived from pancreatic juice (green circles) (Figure 4B). The expression data were then subjected to an unsupervised hierarchical clustering resulting in a heatmap where the samples belonging to the pancreatic cancer patients (both from blood and pancreatic drainage) were clustered together against the control group (Figure 4C). We then compared the microRNA expression from the samples derived from blood and pancreatic juice individually against the control group, obtaining a clear discriminatory result from cancer vs. control in each case (Figure 4D top and middle panel). Interestingly, when we compared the microRNA expression of platelets obtained from cancer patients´ blood versus those obtained from pancreatic juice, we found that they differed by clustering separately (Figure 4D bottom panel). These results suggest the possible presence of platelets’ subpopulations of platelets, as defined by the specific expression of selected microRNAs. A larger number of samples would be needed to address this, since the unsupervised hierarchical clustering using all of the samples was unable to corroborate this, even with the PCA and separate analysis.

We then selected the top eight deregulated microRNAs (*p* < 0.0001) (mir-181a-3p, mir-151b, mir-711, mir-624, mir-2115-5p, mir-600, mir-6820 and mir-4632-5p) as potential candidates for a pancreatic cancer platelet microRNA genomic signature since every one of these miRNAs were found differentially expressed against the control samples in all of the biological sources. We analyzed the expression of four of these microRNAs in platelets co-cultured with pancreatic cancer cells (Figure 4E). We were able to validate the microarray expression changes and tendencies of these microRNAs in our trained platelets.

## 3. Discussion

Pancreatic tumors are silent, and they are usually detected at very advanced stages, which reflects in the low opportunities for treatment and, thus, a high mortality rate. To improve this, several approaches have been envisioned, most of them aimed toward the use of new blood markers, including the so-called “liquid biopsies”. Liquid biopsies have several advantages over simple biochemical or proteomic approaches, since they provide more information that can be used not only for cancer detection, but also for its classification, prognosis or even for therapeutic approaches. Nevertheless, the main problem is the sensitivity, which is low due to the amount of tumoral cells or free cancer-derived nucleic acids in the blood. Platelets could be an excellent surrogate liquid biopsy, acting also as a non-invasive diagnostic test in cancer that is able to amplify the cancer “signal” in the blood [11,13]. It has been shown that the mRNA expression profile differs significantly between the platelets derived from cancer patients and healthy donors, and even among patients with different tumors [32]. Nevertheless, little is known about the molecular mechanisms that mediate these differences. As platelets are non-nucleated, the steady-state RNA levels are regulated by post-transcriptional mechanisms, including microRNA-mediated mechanisms. This makes microRNA the prime candidates as mediators of the RNA expression differences in platelets derived from cancer patients. Each microRNA is able to regulate one to hundreds of mRNAs [45], so they constitute gene expression control hubs. Even discrete changes in these molecules can have dramatic effects on multiple signaling networks [46]. Since platelets have the whole cellular machinery needed to process and regulate miRNAs, the association of these cells with cancer cells can explain multiple processes present in the tumor microenvironment [47].

Very little is known about the interaction of platelets and pancreatic cancer cells. In this work we found that the co-culture of platelets and cancer cells induced a phenotypic change of BxPC-3 cells with a concomitant change in the miRNA expression profile of platelets. Platelet education is still a new and poorly described process. In the present report, we demonstrated that platelets from healthy donors or derived in vitro can be educated when exposed to pancreatic cancer cells. This opens the door for more insightful research on the nature and regulation of this process. The co-culture of platelets and cancer cells causes the latter to increment their proliferation rate as well as to increase the ability to migrate and form colonies in soft agar. These results indicate the acquisition of a more aggressive phenotype, which is accompanied by an expansion in the number of stem cells, as determined by an ELDA analysis. In accordance, we found a clear overexpression of the genes associated with the stem phenotype, such as Oct4, SOX2 and NANOG. It has been reported that platelets play a physiological role in the modulation of the immune system, by regulating local inflammatory responses, secreting several proliferative cytokines and even regulating processes such as the epithelium–mesenchymal transition [48,49,50]. MicroRNAs, as key gene regulatory hubs, may be involved in the effects elicited by educated platelets. Further studies using gain- and loss-of-function should help to clarify this.

In the present report, we found differences in the expression of several platelet microRNAs. These differences were enough to separate the control and cancer patients in an unsupervised analysis. Interestingly, a PCA and a separate unsupervised clustering analysis also separated the platelets derived from blood and pancreatic juice when the same patient was subjected to the analysis. Clustering was not preserved when the healthy controls were considered, perhaps due to the low number of samples analyzed, as the platelets derived from blood and pancreatic juice from cancer patients were more similar. These results point toward the presence of a subset population of platelets in the same patient. We currently do not know the reason for these differences, but the effect of postsurgical inflammation, lack of blood components in the pancreatic juice or even the possible effect of pancreatic cancer in the region should be considered. A comparison of platelets derived from pancreatic juice obtained by other methods (e.g., endoscopy) and other diseases could be helpful to determine the origin of these changes. Supporting the possibility that the platelets subpopulation is indeed composed of educated platelets, we found that four of the differentially expressed ´miRNAs (mir-600, mir-711, mir-6865, mir-2115, mir-624-5p) were also present in the blood of pancreatic cancer patients. These platelets are accessible from the blood and could be used as a non-invasive pancreatic cancer biomarker.

There is still a lot to be discovered about the platelet education process and its relevance as a biological phenomenon and clinical utility. Some authors are already calling the platelets as the “Holy Grail” of cancer diagnosis [51], but more caution and research is needed until more data are produced to support this statement. Since pancreatitis is a known risk factor for pancreatic adenocarcinoma, a similar number of platelets and microRNA expression cassette could be expected. Nevertheless, it has been documented that this is not the case, as shown by Kefeli et al. in 2014 [52]. In this report, the platelet number in active and in remission pancreatitis patients was statistically lower than that of control subjects. This result clearly contrasts with previous reports of thrombocytosis found in pancreatic cancer patients. Likewise, we did not find coincidences between the DE miRNAs that we found in the present article and the DE miRNAs that were reported in the literature [53]. We hope to expand the number of clinical samples and provide a possible cellular mechanism for the effects of the described microRNAs in pancreatic cancer development to fulfill or reject these expectations.

## 4. Materials and Methods

**Patient samples.** The research protocol was approved by the Research and Ethics Committees of the Instituto Nacional de la Nutricion Salvador Zubiran (INCMNSZ) and the Instituto Nacional de Medicina Genomica. Informed consent was obtained from all of the patients. In total, 24 patients were sampled. All of the patients were informed of the risks and possible complications of participating in the study, as well as their contribution to pancreatic cancer research in the Mexican population. After an interview, where all the doubts that patients might have been resolved, the informed consent was signed in accordance with the guidelines of the Ethics Commission of the INCMNSZ. Consent was obtained one day before the surgery and the samples were collected according to the following workflow: first day (pre-surgical) 10 mL of peripheral blood obtained, third day (postsurgical) pancreatic fluid obtained from the surgical drainage.

**Isolation of blood-derived platelets.** The collected samples were taken from the peripheral vein of the arm by means of venipuncture by the nursing staff responsible for the patient. Blood was collected in blue-capped Vacutainer tubes, which contained sodium citrate 0.109 M (citrate 3.2%). Then, the platelet-rich plasma (PRP) was obtained with the following methodology: (1) The tubes were centrifuged at 120× *g* rcf for 10 min at room temperature; (2) The top 2/3 of the obtained supernatant was taken, which is platelet-rich plasma (PRP) and was transferred to a new tube; (3) The lower third was discarded to avoid contamination with cells of the white fraction; (4) HEPES Buffer with 200 mM of acetylsalicylic acid in a proportion of 1:1 was added to the PRP; (5) The tube was incubated at room temperature for 10 min and centrifuged to 1000× *g* rcf for 15 min at room temperature; the supernatant was removed again to remove traces of red or white cells; (6) The supernatant was again centrifuged at 100× *g* rcf for 20 min at room temperature; (7) The resulting cellular button was considered to be purified platelets.

**Isolation of platelets derived from pancreatic juice.** Postsurgical pancreatic juice samples from the Penrose drainage tube were collected 1 day after surgery from patients, in sterile Falcon tubes. The samples were then processed using the same protocol described for the blood platelets. Once the platelets were obtained from pancreatic juice samples, RNA was extracted from them using Trizol Reagent (Thermo Fischer cat#15596026, Waltham, MA USA) according to the manufacturer´s instructions and frozen at −70 °C until used.

**Analysis of platelets microRNAs using microarrays.** Platelet RNA from pancreatic cancer patients and healthy individuals was used for a microRNAs differential expression assay using Affymetrix microarrays (miRNA-4_0-st-v1 version). A total of 12 samples of platelet microRNAs were analyzed using this platform. These samples were divided in 3 different biological groups: (1) A total of 4 blood samples from the patient’s blood; (2) A total of 4 pancreatic juice samples derived from post-surgical drainage; and (3) A total of 4 blood samples from healthy individuals. The samples from healthy donors were considered the control group. The results of the microarrays were analyzed through various platforms: R studio (oligo, EDGE, Deseq2 packages), GenomicScape and MORPHEUS (noise-signal analysis) from the Broad Institute [54,55,56,57]. Data were deposited at the NIH´s Gene Expression Omnibus (GSE203122).

**RNA extraction and real-time quantitative PCR (qPCR) analysis.** Total RNA was extracted using Invitrogen Trizol Reagent (Thermo Fischer cat#15596026, Waltham, MA USA) according to the manufacturer´s instructions. The quality and quantity of the RNA were determined by a Nanodrop (Thermo Scientific ND1000, Waltham, MA USA). A total of 200 ng of RNA were used to generate cDNA with the TaqMan Advanced miRNA cDNA Synthesis kit (Applied Biosystems cat# A25576, Waltham, MA USA). mRNA cDNA was synthesized with the Maxima First Strand cDNA Synthesis Kit for RT-qPCR (Thermo Fisher, #cat K1641, Waltham, MA USA). The qPCR was run in the QuantStudio™ 7 Flex Real-Time PCR System (Thermo Fisher 4485701, Waltham, MA USA). Relative quantification of RT-PCR was determined by the amplification efficiency and the cycle number at which fluorescence crossed a pre-scribed background level cycle threshold (Ct). The relative expression levels of miRNAs were calculated by 2−△CT method (the sequences of the primers utilized can be consulted in Appendix A).

**Cell culture.** The BxPC-3 cell line, derived from a pancreatic adenocarcinoma (ATCC^®^ CRL-1687™, Manassas VA, USA) was cultured RPMI 1640 medium (ATCC, Manassas, VA, USA) supplemented with L-glutamine, 10% of fetal bovine serum (FBS; SIGMA-ALDRICH; St. Lois, MO, USA) at 37 °C and 5% CO_2_ in monolayer cultures. The megakaryoblastic cell line MEG-01 (ATCC^®^ CRL-2021^TM^; Manassas, VA, USA) was grown in suspension with RPMI 1640, supplemented with L-glutamine, 10% fetal bovine serum (FBS; SIGMA-ALDRICH; St. Lois, MO, USA) at 37 °C and 5% CO_2_.

**Non-contact platelet training.** This was performed on 75 mm Transwells^®^ with 3μm pores (Corning, cat#3420, Corning, NY, USA) to prevent cells from passing through them. Since the platelets have an average size of 3 μm, these were seeded at the bottom of the base plate, which was coated with poly-L-Lysine (Sigma-Aldrich, cat# P8920, St. Louis, MO, USA). Subsequently, the insert was seeded with 30,000 BxPC-3 cells. RPMI medium was used for all of the experiments. The platelets were collected 72 h later, and RNA was extracted from them as previously described.

**Platelet production from MEG-01 cells.** To induce a high and synchronized production of platelets, we used the MEG-01 megakaryoblastic cell line, which can differentiate into megakaryocytic-like cells to produce platelets. Human thrombopoietin (TPO, Sigma-Aldrich, #cat SRP3178, St. Lois, MO, USA) and valproic acid (VA, Sigma-Aldrich, #cat P4543, St. Louis, MO, USA) were added to MEG-01 cells to this end. TPO was used at a concentration of 10 ng/mL, whereas AV was used at 2 mM concentration. In the case where both (TPO, AV) inductors were present, the same concentrations were used, respectively.

**Purification of platelets from MEG-01 cells.** Platelet-like particles (PLP) were obtained from MEG-01 cultures treated with TPO or VA at 2 mM concentration for 48 h to 37 °C. The supernatants of these cultures were collected and centrifuged to 120× *g* for 10 min. The obtained supernatants were centrifugated again to 1000× *g* for 15 min. After this centrifugation, the pellet containing the PLP was collected.

**Clot Formation Assay.** Isolated platelets from patients or differentiated MEG-01 cells were collected, washed and resuspended in PBS. CaCl_2_ was added to each tube for a final concentration of 50 mM. After 10 min at room temperature, the tubes with CaCl_2_ formed a thick clot.

**Immunofluorescence assay.** BxPC-3 cells (250,000) were seeded on top of sterile coverslips. After 24 h, 37 million platelets collected from healthy controls were added into each well (150 platelets/BxPC-3 cell) and incubated for 72 h at 37 °C and 5% of CO_2_. After incubation, the cells were gently washed with PBS once and fixed using 4% paraformaldehyde for 20 min at room temperature. After fixation, each well was washed twice with PBS and a 2.5 µM DAPI solution (Thermo Fisher cat# 62248, Waltham, MA USA) was added and incubated in the dark at room temperature for 20 min. The DAPI solution was washed twice with PBS and an antibody for CD41 (abcam cat#ab134131, Cambridge, UK) was added at a 1:100 dilution and incubated for further 30 min at room temperature. Next, the cells were washed once again with PBS twice and the secondary antibody at a 1:1000 dilution (Abcam cat # ab150077, Cambridge, UK) was added and incubated for 30 min. Cells were finally washed with PBS twice and FM 4-64 (Thermo Fisher cat # T13320, Waltham, MA, USA) was added and incubated at 4 °C for 10 min. Cells were quickly washed with PBS and visualized immediately by fluorescence microscopy (Axiostar Plus, ZEISS, GE).

**Flow Cytometry.** The platelets were marked with an anti-CD41 antibody for 30 min and a secondary Alexa Fluor-488 antibody for an additional 30 min. Flow cytometry was performed utilizing a FACSMelody Cell Sorter (BD Biosciencies, Franklin Lanes, NJ, USA).

**Migration Assays.** These assays were conducted using Transwells^®^ well inserts of 0.47 cm2 area with 8 microns pore (Thermo Fisher, # cat 140629, Waltham, MA USA). 35,000 BxPC-3 (control vs. cell pre-incubated with platelets) were seeded at the top of the insert and RPMI supplemented with 10% FBS was used as chemoattractant. Cells were collected at 24 h, 48 h, and 72 h, fixed with a solution of 4% paraformaldehyde and stained with Crystal violet 0.05%. Statistical analyses were made using a Welch’s *t*-test (*p* < 0.001) analysis in GraphPad Prism 9 software.

**Extreme Limiting Dilution Analysis (ELDA).** Extreme Limiting Dilution Analyses were conducted on 96-well plates by sequential dilutions starting with 10,000 cells. Plates were left in incubation at 37 °C and 5% de CO_2_ for 7 days, adding RPMI every fourth day to avoid evaporation. Data were analyzed and graph by R’s limdil (ELDA) [43] function of the statmod package (ver. 1.4.36).

## 5. Conclusions 

Our results show that a co-culture of platelets and pancreatic cancer cells induce important changes in both cell types. In cancer cells, the capacity for migration, proliferation, clonogenicity and stemness increases, with a clear change in the expression of Nanog, Oct4 and SOX2 transcription factors. The platelets from pancreatic cancer patients showed differences in their microRNA profile when compared to the platelets from individuals without cancer. Interestingly, we detected a possible platelet subpopulation present in the pancreatic juice derived from postsurgical pancreatic cancer patients, as assessed by their microRNAs expression profile.

## Figures and Tables

**Figure 1 ijms-23-11438-f001:**
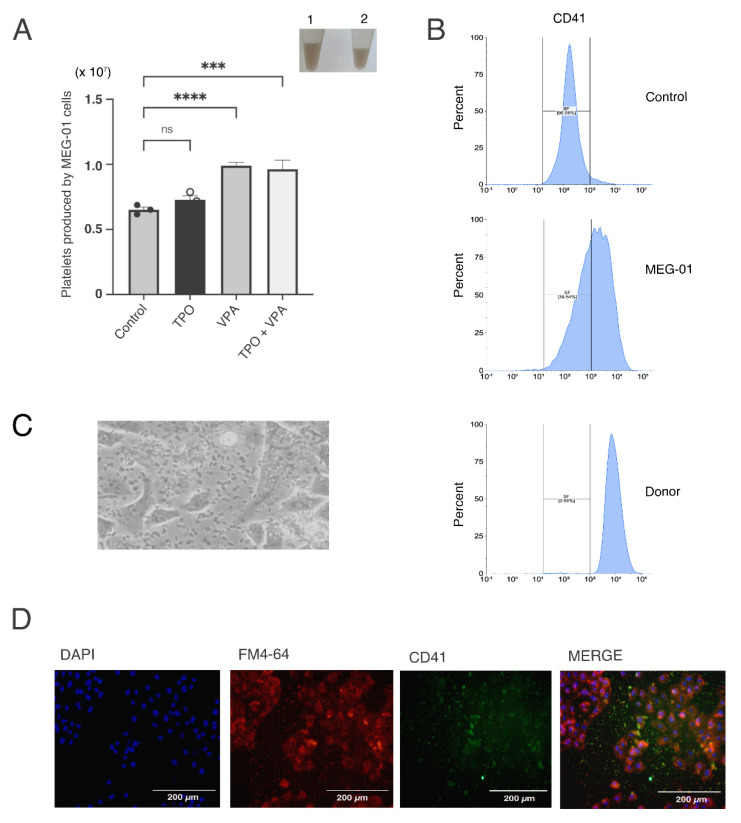
In vitro platelet production by different inductors. (**A**) MEG−01-derived platelets were counted after 72 h of treatment with: VPA (acid valproic), TPO (thrombopoietin) and a combination of both. Platelets were manually counted in a Neubauer chamber. The asterisks indicate a *p* < 0.005. Inset: Left: Photograph of CaCl_2_-induced clot formed by platelets co-cultured with BxPC−3 cancer cells (left panel, 1) or MEG−01-derived platelets; (ns: non-stained platelets) (**B**) Representative flow cytometry histogram of CD41-labelled platelets. Top panel: control (ns: non-stained platelets) isolated from healthy donors; middle panel: platelets produced by MEG-01 differentiation and lower panel: platelets isolated from healthy donors; (**C**) Photograph of platelets co-cultured with BxPC−3 cancer cells; (**D**) Immunofluorescence of BxPC−3 cells in co-culture with platelets. CD41 was stained with AlexaFlour488 (green), membranes were stained with FM 4-64 dye (red) and their nuclei were stained with DAPI (blue). *** *p* < 0.001, **** *p* < 0.0001.

**Figure 2 ijms-23-11438-f002:**
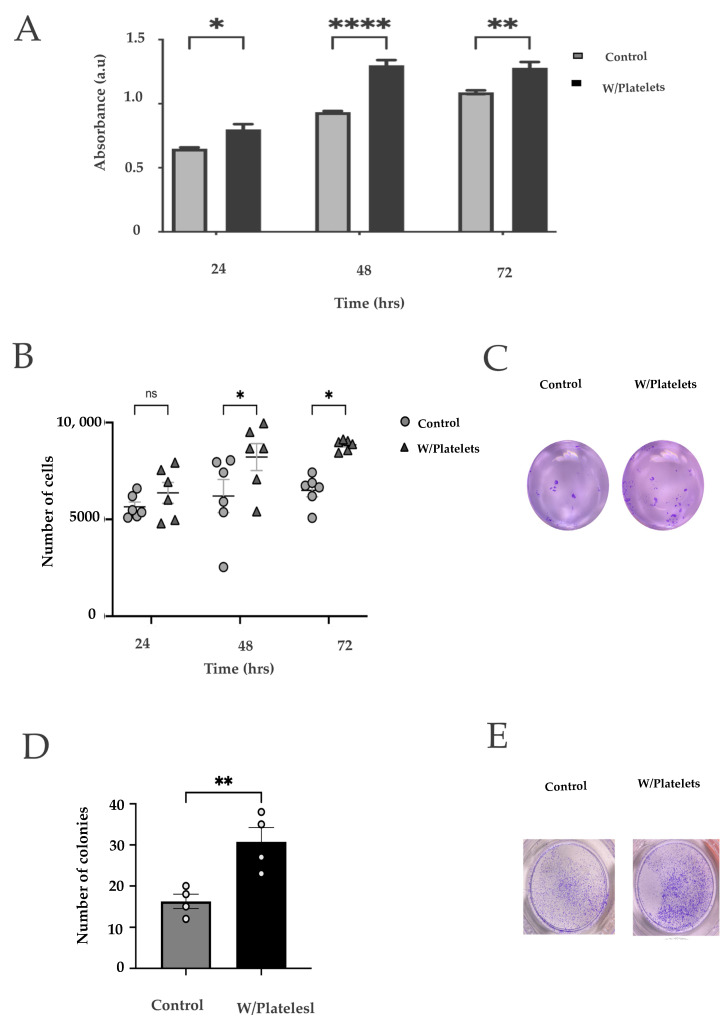
**Effects of the platelets on cancer cells**. (**A**) Viability curves of BxPC−3 cells exposed to platelets for 24, 48, and 72 h using the MTT assay; (ns: non-stained platelets) (**B**) Clonogenicity assay; 9 serial dilutions of cells were made, cultured for two weeks, and stained with crystal violet. Both groups were observed and quantified at 3 time points: 24 h, 48 h and 72 h; (**C**) Representative clonogenicity assays of unexposed cells (left) or cells exposed to platelets (right) after 7 days of culture on a double layer of semi-solid agar [40]; (**D**) Migration of BxPC−3 cells exposed and not exposed to platelets for 48 hrs; (**E**) Example image of a migration assay of BxPC−3 cells exposed to platelets. ns: not significant, * *p* < 0.05, ** *p* < 0.01, **** *p* < 0.0001.

**Figure 3 ijms-23-11438-f003:**
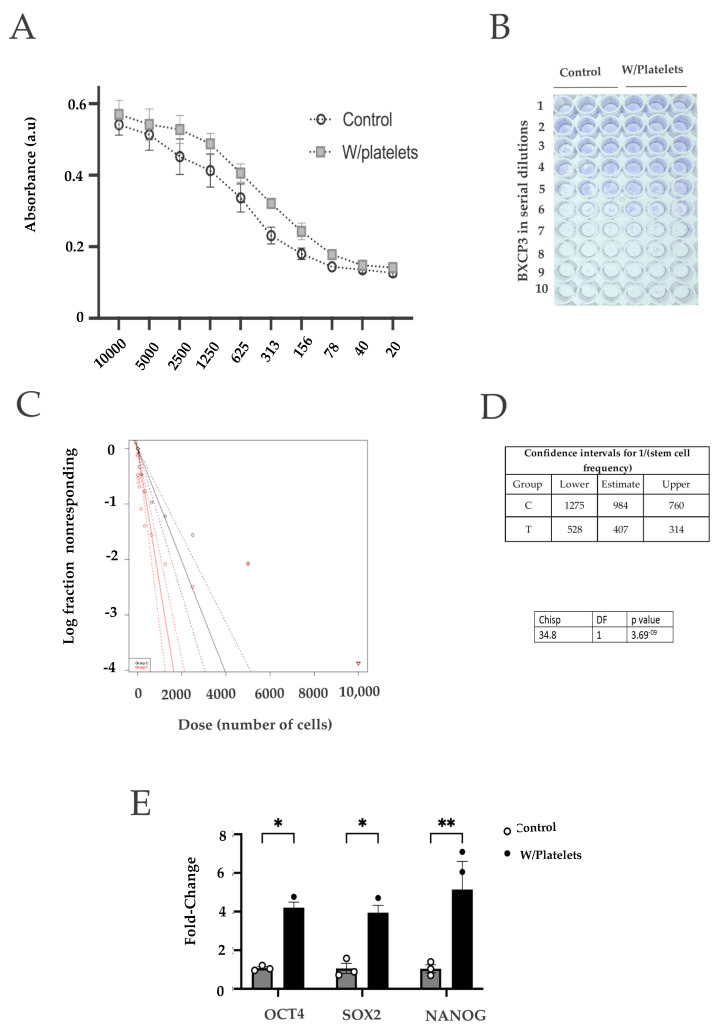
**Cancer cells exposed to platelets present stem-like characteristics.** (**A**) Clonogenicity assay using 9 serial dilutions of ΒxPC−3 cells exposed to platelets; (**B**) and (**C**) Calculation of the proportion of stem cells present in the BxPC−3 culture by the ELDA method with (red) or without (black) platelets. The presence of platelets increases the ratio of stem cells of BxPC−3 cells about 10 times more than in control cells; (**D**) qPCR quantification of 3 genes directly associated with the self-renewal and pluripotent state characteristic of cancer stem cells. * *p* < 0.05, ** *p* < 0.01.

**Figure 4 ijms-23-11438-f004:**
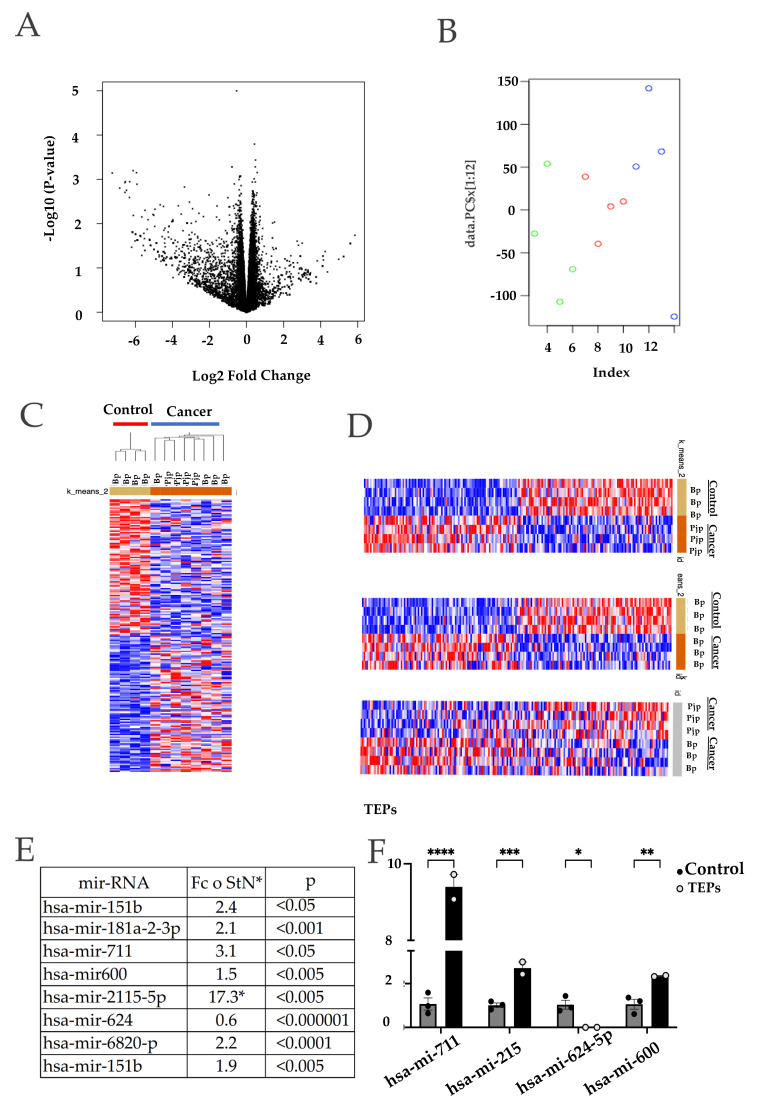
**Differentially expressed miRNAs in platelets derived from pancreatic cancer patients (Tumor Educated Platelets).** A total of 4 blood samples and 4 pancreatic juice samples were taken from 4 different patients of pancreatic cancer. These samples were paired with 4 blood samples from healthy donors. (**A**) Volcano plot that shows statistical (*p* value) versus magnitude of change (fold change) of differentially expressed platelet transcriptome of patients with tumor cancer and of people without cancer; (**B**) Principal component analysis of platelets’ transcriptome of patients with tumor cancer and of people without cancer; (**C**) Unsupervised hierarchical clustering of all samples. Two groups were found that discriminates between pancreatic cancer and control individuals: (**D**) Clustering heatmap of pancreatic drainage samples vs. control blood. Top panel: Two groups were found by k-means clustering algorithm. Heatmap of blood samples cancer vs. control by k-means clustering middle panel. Heat map of platelets from blood (Bp) or platelets of pancreatic juice (Pjp) cancer; (**E**) Result of the machine-learning signal-noise analysis between cancer samples, in this case we selected the top miRNAs to 8 candidates with an optimal *p*-value; (**F**) Top 4 differentially expressed miRNAs quantification by qPCR from platelets trained with direct contact with the BxPC-3 cells against control platelets (* *p* < 0.05, ** *p* < 0.01, *** *p* < 0.001, **** *p* < 0.0001).

## Data Availability

Microarray data were deposited at the NIH’s Gene Expression Omnibus GSE203122 Avalible online: https://www.ncbi.nlm.nih.gov/geo/ (accessed on 20 may 2022).

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
