# Peer review of "Pancreatic Cancer Cells Induce MicroRNA Deregulation in Platelets"

_ijms, 2022, doi:10.3390/ijms231911438_

Round 1

Reviewer 1 Report

Line 37 Typo: replace has for as. The typo is repeated twice in the same line.

Line 40 instead of late stage it would be more appropriate to say advanced stage. The stage is not late, it is advanced. What is late is the diagnosis.

Line 42 replace explain for explained

Line 43 change syntax

Line 44 replace injure for lesion

Line 54 replace get for obtain

Line 57 please revise your definition of platelets. Non-nucleated should replace enucleated. Platelets are not enucleated.

Line 67 idem.

Line 74 Replace loadout for load.

Lines 87-88 improve syntax

Line 88 delete E, only leave the surname otherwise write the full name.

Line 90 idem delete O

Line 91 the word >effect> is used three times in the same sentence. Improve syntax

Line 160 express should be expressed.

Line 185 correct enucleated

Line 194 to 197 is confusing. Rewrite it in an understandable manner.

Line 259 correct enucleated

The main issue that lacks in this paper is the study of patients with pancreatitis. The fact that there is a different expression of MiRNAs between controls and PDAC patients will have real value if pancreatitis patients do not present the same signature as PDAC.

I think this issue should be commented in the paper.

Author Response

                                                                                                          September 10th 2022

Response letter to reviewers

Journal: International Journal of Molecular Sciences

Manuscript ID: ijms-1924363

Title: Article

Pancreatic Cancer Cells induce microRNA deregulation in platelets

Authors: Jorge Yassen Díaz-Blancas, Ismael Dominguez-Rosado, Carlos

Chan-Nuñez, Jorge Melendez-Zajgla, Vilma Maldonado *

We appreciate in advance the valuable and pertinent contributions of the reviewers to the manuscript that helped to improve it and clarify various points.

Reviewer 1.

Line 37 Typo: replace has for as. The typo is repeated twice in the same line.

R= We corrected this in the last version.

Line 40 instead of late stage it would be more appropriate to say advanced stage. The stage is not late, it is advanced. What is late is the diagnosis.

R= We corrected this in the last version.

Line 42 replace explain for explained

R= We corrected this in the last version

Line 43 change syntax

R= We corrected this in the last version

Line 44 replace injure for lesion

R= We corrected this in the last version

Line 54 replace get for obtain

R= We corrected this in the last version

Line 57 please revise your definition of platelets. Non-nucleated should replace enucleated. Platelets are not enucleated.

R= We corrected this in the last version

Line 54 replace get for obtain

R= We corrected this in the last version

Line 67 idem.

R= We corrected this in the last version

Line 74 Replace loadout for load.

R= We corrected this in the last version

Lines 87-88 improve syntax

R= We corrected this in the last version

Line 88 delete E, only leave the surname otherwise write the full name.

R= We corrected this in the last version

Line 90 idem delete O

R= We corrected this in the last version

Line 91 the word >effect> is used three times in the same sentence. Improve syntax

R= We corrected this in the last version

Line 160 express should be expressed.

R= We corrected this in the last version

Line 185 correct enucleated

R= We corrected this in the last version

Line 194 to 197 is confusing. Rewrite it in an understandable manner.

R= We corrected this in the last version

Line 259 correct enucleated

R= We corrected this in the last version

The main issue that lacks in this paper is the study of patients with pancreatitis. The fact that there is a different expression of MiRNAs between controls and PDAC patients will have real value if pancreatitis patients do not present the same signature as PDAC.

I think this issue should be commented in the paper.

R= We added a paragraph in the discussion to clarify the point

R= Since pancreatitis is a known risk factor for pancreatic adenocarcinoma, a similar platelets´number and microRNA expression could be expected. Nevertheless, it has been documented that this is not the case, as shown by Kefeli et. al. in 2014 (Euroasian Journal of Hepato-Gastroenterology, July-December 2014;4(2):67-69). In this analysis, the platelet number in active and in remission pancreatitis patients was statistically lower than that of control subjects. This result clearly contrasts to previous reports of thrombocytosis found in pancreatic cancer patients. Likewise, we did not find coincidences between the DE miRNAs we found in the present article and the DE miRNAs that have been reported in the literature (MicroRNAs in acute pancreatitis: From pathogenesis to novel diagnosis and therap, J Cell Physiol. 2019;1–14).

.

Reviewer 2 Report

This study aims at a conclusion that “Pancreatic Cancer Cells induce microRNA deregulation in platelets”, and indeed, the authors have observed an interesting dysregulation of microRNA expression in pancreatic cancer patients (Figure 4). However, in rest of the paper, the authors focused on the effects of the platelets on cancer cells (Figure 1-3). Platelets may have an important effect on cancer cells, but this evidence is not enough to support the aimed conclusion. To claim “Pancreatic Cancer Cells induce microRNA deregulation in platelets”, it is essential to show that platelets microRNA expressions are changed when co-cultured with cancer cells. Current manuscript is not ready for publish as this important piece of data is still missing.

Minor comments:

1. Line 18, “early-stage diagnostic biomarkers is essential”. Are the pancreatic cancer patients included in this study at the early-stage for diagnostic?

2. Line 25-26, “we also determined the differential expression of miRNAs in platelets obtained from a small cohort of pancreatic cancer patients.” Line 231-231, “Heat map of platelets from blood (Bp) or platelets of pancreatic juice (Pjp) cancer”. Line 350-351, “Platelet RNA from pancreatic cancer patients and healthy individuals was used for..” This information is conflicting, are the platelets comparisons done between blood and pancreatic juice (Pjp) cancer sample from same patents, or between cancer patients and healthy individuals?

3. Line 156, “Migration of BxPC-3 or control cells exposed to platelets for 48 hrs”. What are the control cells, another cell line? Or should it be “Migration of BxPC-3 cells exposed/not exposed to platelets?

4. Figure 4E is not readable.

Author Response

                                                                                                          September 10th 2022

Response letter to reviewers

Journal: International Journal of Molecular Sciences

Manuscript ID: ijms-1924363

Title: Article

Pancreatic Cancer Cells induce microRNA deregulation in platelets

Authors: Jorge Yassen Díaz-Blancas, Ismael Dominguez-Rosado, Carlos

Chan-Nuñez, Jorge Melendez-Zajgla, Vilma Maldonado *

We appreciate in advance the valuable and pertinent contributions of the reviewers to the manuscript that helped to improve it and clarify various points. 

Reviewer 2

This study aims at a conclusion that “Pancreatic Cancer Cells induce microRNA deregulation in platelets”, and indeed, the authors have observed an interesting dysregulation of microRNA expression in pancreatic cancer patients (Figure 4). However, in rest of the paper, the authors focused on the effects of the platelets on cancer cells (Figure 1-3). Platelets may have an important effect on cancer cells, but this evidence is not enough to support the aimed conclusion. To claim “Pancreatic Cancer Cells induce microRNA deregulation in platelets”, it is essential to show that platelets microRNA expressions are changed when co-cultured with cancer cells. Current manuscript is not ready for publish as this important piece of data is still missing.

R= In Figure 4 subsection F, these results are shown (Top 4 quantification of miRNAs differentially expressed by qPCR from platelets trained with direct contact with BxPC-3 cells against control platelets), the figure was corrected, and subsection (F) was correctly marked in the figure 4.

Minor comments:

  1. Line 18, “early-stage diagnostic biomarkers is essential”. Are the pancreatic cancer patients included in this study at the early-stage for diagnostic?

R= Unfortunately, it is very rare that pancreatic cancer patients are diagnosed in the early stage. Symptoms are usually associated with later stages, and this is one of the reasons for the bad prognosis in pancreatic cancer. Nevertheless, we are assuming that the differences in microRNAs expression are maintained between early and late stage. This, of course, has to be explored and we are now starting a new study looking for the expression of 4-6 miRNAs in platelets as a liquid biopsy (non-invasive) in patients with suspected pancreatic cancer. We hope that we can obtain at least some early stage patients for this.

  1. Line 25-26, “we also determined the differential expression of miRNAs in platelets obtained from a small cohort of pancreatic cancer patients.” Line 231-231, “Heat map of platelets from blood (Bp) or platelets of pancreatic juice (Pjp) cancer”. Line 350-351, “Platelet RNA from pancreatic cancer patients and healthy individuals was used for..” This information is conflicting, are the platelets comparisons done between blood and pancreatic juice (Pjp) cance sample from same patents, or between cancer patients and healthy individuals?

R=Thank you for pointing this out. In line 25-26 we added that we also analyzed samples of healthy individuals in the study. Two comparisons were made: platelets obtained from the blood of healthy individuals with those obtained from cancer patients, and an analysis comparing platelets obtained from blood cells and pancreatic juice from the same patients (paired analysis). With this, we are assuming that the pancreatic juice had platelets that were in more direct contact with the tumor and that it could have different miRNAs to those in the general blood pool. We have improved the wording in the manuscript to clarify this topic.

  1. Line 156, “Migration of BxPC-3 or control cells exposed to platelets for 48 hrs”. What are the control cells, another cell line? Or should it be “Migration?

R= We corrected this in the last version

  1. Figure 4E is not readable.

R= We corrected this in the last version

Round 2

Reviewer 2 Report

Issues raised in my review were addressed by the authors. Thank you.